# Relationships between Child Development at School Entry and Adolescent Health—A Participatory Systematic Review

**DOI:** 10.3390/ijerph182111613

**Published:** 2021-11-04

**Authors:** Michelle Black, Amy Barnes, Mark Strong, Anna Brook, Anna Ray, Ben Holden, Clare Foster, David Taylor-Robinson

**Affiliations:** 1School of Health and Related Research, The University of Sheffield, Regent Court, 30 Regent Street, Sheffield S1 4DA, UK; a.barnes@sheffield.ac.uk (A.B.); m.strong@sheffield.ac.uk (M.S.); anna.brook@sheffield.ac.uk (A.B.); b.holden@sheffield.ac.uk (B.H.); clare.foster@sheffield.ac.uk (C.F.); 2Department of Health Sciences, University of York, Seebohm Rowntree Building, Heslington, York YO10 5DD, UK; annamarie.ray@nhs.net; 3Public Health, Policy and Systems, Institute of Population Health, University of Liverpool, Liverpool L69 3GL, UK; David.Taylor-Robinson@liverpool.ac.uk

**Keywords:** child development, childhood education, school, adolescent health, health inequality, adolescent mental health, adolescent weight

## Abstract

The relationship between child development and adolescent health, and how this may be modified by socio-economic conditions, is poorly understood. This limits cross-sector interventions to address adolescent health inequality. This review summarises evidence on the associations between child development at school starting age and subsequent health in adolescence and identifies factors affecting associations. We undertook a participatory systematic review, searching electronic databases (MEDLINE, PsycINFO, ASSIA and ERIC) for articles published between November 1990 and November 2020. Observational, intervention and review studies reporting a measure of child development and subsequent health outcomes, specifically weight and mental health, were included. Studies were individually and collectively assessed for quality using a comparative rating system of stronger, weaker, inconsistent or limited evidence. Associations between child development and adolescent health outcomes were assessed and reported by four domains of child development (socio-emotional, cognitive, language and communication, and physical development). A conceptual diagram, produced with stakeholders at the outset of the study, acted as a framework for narrative synthesis of factors that modify or mediate associations. Thirty-four studies were included. Analysis indicated stronger evidence of associations between measures of socio-emotional development and subsequent mental health and weight outcomes; in particular, positive associations between early externalising behaviours and later internalising and externalising, and negative associations between emotional wellbeing and later internalising and unhealthy weight. For all other domains of child development, although associations with subsequent health were positive, the evidence was either weaker, inconsistent or limited. There was limited evidence on factors that altered associations. Positive socio-emotional development at school starting age appears particularly important for subsequent mental health and weight in adolescence. More collaborative research across health and education is needed on other domains of development and on the mechanisms that link development and later health, and on how any relationship is modified by socio-economic context.

## 1. Introduction

Inequalities in many child health outcomes are increasing in the UK and the health of those living in its most disadvantaged areas are amongst the worst in the developed world [1]. Some of the roots of health inequality are thought to be in early childhood with socio-economically driven inequalities in child development persisting across the life course, negatively impacting people’s future health, wellbeing and life chances, and perpetuating health inequalities into adulthood [2]. Evidence that the early years, or the first “1000 days”, is a critical period of development [3,4] (together with health economics research in this field [5]) has meant that the early years have become a prime area for public policy and public health investment in many high-income countries including the UK [6].

All of the countries of the UK provide early childhood programmes, which aim to improve outcomes for children by supporting optimal health and development through access to services such as early education and care, between the ages of 0–4 years or pre-school [7]. There is evidence that programmes which support child development in readiness for school can improve cognitive and non-cognitive skills [8]. There is also evidence that positive cognitive development on starting school is associated with academic achievement by age 13 years [9] and positive socio-emotional development by age 10 years [10]. Non-cognitive skills, such as social skills and self-regulation on starting school, are also associated with later academic success and psychosocial outcomes in subsequent years of childhood and early adolescence [11]. There is less evidence for whether and how child development, or interventions to support child development, are related to subsequent health in childhood. For example, there is limited evidence on the effect of early child development programmes (such as attending pre-school, accessing health services and parenting programmes) on adolescent health, with one systematic review finding little to no effect of early childhood programmes on later child health, although with some evidence for obesity reduction, greater social competence, improved mental health and crime prevention [12]. A review of Sure Start (a UK early years programme from 1999 to 2017, for families with children under the age of four years and targeted in more disadvantaged areas) found that access to Sure Start was associated with fewer childhood hospitalisations for infections and injury [13]. Potential mechanisms proposed for this association were: the provision of information to parents and changing parents behaviour, leading to a safer and more nurturing home, and to reducing externalising behaviour in children, leading to less fights or dangerous activities [13].

To better understand whether and how child development at school starting age is associated with subsequent health in childhood requires a clear understanding of what is meant by “child development”, reliable measures of child development, and also the development and testing of conceptual frameworks or theories regarding the relationships between child development and later adolescent health. In terms of defining what we mean by child development, this is contested academic and policy terrain and, as such, is difficult to define. For some, child development is understood through a narrow focus on cognitive education, whereas for others it is about broader life skills, including confidence and social competencies [14]. In English health and education policy, child development has tended to be defined in the former, relatively narrow manner, with, for example, child development at school starting age understood through a specific composite measure of a child’s personal, social and emotional, physical, cognitive, and communication and language development, termed “school readiness” [15]. Internationally, school readiness, when considered more broadly, has been seen as a viable strategy to reduce inequalities in learning and development gaps at the start of formal education [16]. However, how it is defined and used in England has been criticised as reductionist, with school readiness used as a performance and accountability measure, resulting in a narrowing of the curriculum, marginalisation of children who fail to achieve required levels of development through grouping by ability, and subjugation of teachers and schools to meet targets [17]. Moving beyond targets to understanding child development more broadly, as an ongoing developmental process in a social context [18], is important if we are to develop interventions to support equitable health and development. Therefore, we consider “child development” in this review as any measure of child development which encapsulates a process of change in what a child is capable or able of doing, or in how they are feeling. There is no existing framework for characterising different aspects or measures of child development. Therefore, in this review we use four over-arching domains of child development: socio-emotional development, cognitive development, language and communication, and physical development. These domains broadly encompass the areas of learning within the early years curriculum in England [15]. We see these categories as potentially useful despite the described shortcomings of England’s composite measure, “school readiness”. Conceptualising child development in this way provides a platform for learning about the relationships between specific domains of child development (using a range of child development measures) and subsequent health. 

Understanding whether and how child development and adolescent health outcomes are related presents opportunities for interventions to improve health and reduce health inequalities at an important time in the life-course, adolescence. There is evidence that health in adolescence is on the causal pathway to socio-economic status (SES) in adulthood by enabling “selection” into education [19]. Therefore, focusing on health in this period is critical to enable children to optimise their subsequent educational outcomes for wellbeing and employment opportunities. Informing interventions requires evidence not just on associations between child development and adolescent health but also on the effect of socio-economic circumstances on any associations found. In our protocol we outlined pathways by which socio-economically driven health inequalities may manifest (family stress, material living circumstances and parental health behaviours) and also possible direct pathways (social and cognitive) between child development and subsequent health. This provides a conceptual framework for the review. To inform interventions on any of these pathways there is a need to identify factors which may explain, and the socio-economic circumstances which may modify, the associations between child development and adolescent health. This requires a public health lens and, as far as we are aware, no review has analysed the evidence on relationships between different dimensions of child development and adolescent health outcomes or assessed the factors which may shape the relationships. 

In summary, there is evidence that aspects of child development at school starting age are associated with later academic success, but less is known about whether and how particular dimensions of child development influence health outcomes in adolescence. This gap in understanding limits cross-sector interventions to improve adolescent health and reduce health inequality. This review addresses this gap by undertaking a participatory systematic review to: (1) synthesise evidence on the relationship between child development at school starting age (3–7 years) and subsequent health in adolescence (8–15 years) and (2) identify factors that shape the relationship. 

## 2. Materials and Methods

### 2.1. Protocol Registration

The study protocol was registered with PROSPERO (CRD42020210011) and published [20]. The review is reported according to the Preferred Reporting Items for Reporting Systematic Reviews and Meta-Analyses (PRISMA) 2020 Statement [21,22]; the checklist available in additional file 1. Any deviations from protocol are stated and explained in the relevant sections.

### 2.2. Review Questions

What are the associations between measures of child development recorded at school starting age (3–7 years) and subsequent health in adolescence (8–15 years)?What are the effect modifiers (socio-economic factors) of this relationship? (This will identify factors which alter the strength of the observed associations.)What are the mediators of this relationship? (This will identify factors or set of factors (pathways) which explain the observed associations.)

### 2.3. Definition of Terms

Child development is defined as a developmental process incorporating measures of development that record changes within a child’s cognitive or physical development, or language and communication, or socio-emotional development.

### 2.4. Study Design

The design for this study was a participatory systematic review, involving engagement with national and local stakeholders across health and education sectors. 

#### Stakeholder Engagement to Design the Conceptual Diagram

The lead reviewer held discussions with stakeholders to develop a conceptual model of the relationship under study. This process is described in full in the study protocol [20]. Their views, together with a scoping review of the evidence, led to an initial conceptual diagram (available in additional file 2). This diagram highlights the main pathways by which socio-economically driven health inequalities manifest; family stress, material living circumstances and parental health behaviours [23]; and also illustrates possible direct pathways (knowledge/literacy and social/cognitive) between child development and education and subsequent health. The diagram acted as a framework for the review, providing initial categories for extracting and analysing evidence from published studies. 

### 2.5. Eligibility

Studies needed to include children, some or all of whom were aged between 3 and 15 years, in high-income country settings defined as a member country of the Organisation for Economic and Co-operation and Development (OECD). Exposures were characteristics of child development at school starting age (3–7 years), defined as: cognitive or physical development, or communication and language, or socio-emotional development. Primary outcomes were health and wellbeing outcomes, reported between the ages of 8 and 15 years: specifically, weight, mental health and proxy measures such as dietary habits and behaviour and measures of wellbeing. Secondary outcomes were academic outcomes of academic tests and proxy measures such as executive function during the outcome age of interest. Secondary outcomes were only included if they were found in a study with a primary outcome of interest. Executive function was included as a secondary outcome of interest because it allows for the regulation, control and management of learning, and thus appears an important link between child development and academic outcomes. In addition, executive function is a good predictor of academic achievement [24]. Studies that provided data on associations between the exposures and outcomes in the age period of interest, and additionally those that provided evidence on mechanisms, were required. The population and context, exposure, outcomes and study designs are described in full in the published protocol [20] and summarised in relation to inclusion and exclusion criteria in Table 1.

### 2.6. Search Strategy

We searched four electronic databases for articles published from November 1990 to November 2020: MEDLINE (OVID), PsycINFO (OVID), ASSIA (ProQuest) and ERIC (EBSCO). We also searched the reference lists from all included articles for additional eligible articles. Further relevant literature was identified through stakeholder discussions. Grey literature searching was undertaken by searching relevant organisations’ websites. The search strategy was informed by a scoping review of the literature and focused on terms relating to child development, school readiness and adolescent health. The search strategy is available in additional file 3. 

### 2.7. Study Selection and Data Extraction

Retrieved citations were uploaded to EndNote and duplicates removed. Titles and abstracts were screened by five reviewers against the inclusion and exclusion criteria. A 10% sample of papers was independently checked by two reviewers and inter-rater reliability was 86%. Any disagreements were resolved by discussion between the reviewers, so that a consensus was reached. The full texts of papers were read in the second stage of the screening process, by five reviewers, to produce a final list of papers for full text review. The final list of papers included was exported to excel to be assessed for the data extraction process. The lead reviewer extracted data for those articles that met the inclusion criteria in full. Reasons for exclusion were recorded and a list of excluded papers, together with the reason, is available in additional file 4. Data extraction was undertaken solely by the lead reviewer using a bespoke form (additional file 5), which had been trialled on a sample of different sources, and a sample of 10% was second checked. The following data were extracted: author and year, study design, analysis method, country and setting, participants, exposure measure and age, exposure measurement instrument, outcome measure and age, outcome measurement instrument, association and effect size, mechanism (studied and proposed), and factors which moderate the association, strengths and weaknesses. 

### 2.8. Quality Assessment

Our protocol stipulated the use of Liverpool Quality Assessment Tool (LQAT) [26]. However, it was found that LQAT was insufficiently detailed for this review. Therefore, in a deviation from protocol we adapted a tool appropriate for the study designs used in previous systematic reviews [27,28]. The methodological quality of each observational study was assessed for risk of bias and clarity of study description to assign studies to one of three categories of methodological quality: high, moderate or low, using the template in additional file 6. Specifically, studies were assessed against 12 criteria within the following categories: study population, study attrition, data collection and data analysis with each pertaining to validity, precision or informativeness. In line with the recommendations of Cochrane [29], studies were not scored, and instead a narrative indication of quality (using +, − and ? against each criteria) was made based on all criteria, with criteria pertaining to validity and precision carrying a greater weight in guiding overall quality. Quality assessments were undertaken by the main author and a 10% sample independently assessed by a member of the review team. In all cases the overall assessments of quality made by the reviewers were consistent. 

In addition to assessing the quality of each individual paper, the overall strength of evidence for papers grouped by outcome and domain was assessed, e.g., mental health outcomes and the socio-emotional domain of child development. Within these groupings the overall findings were graded as providing either: stronger evidence (generally consistent findings in higher quality studies); weaker evidence (generally consistent findings in one higher quality study, or in multiple lower quality studies); inconsistent evidence (inconsistent findings across multiple studies); or very limited evidence (a single study). This method draws on techniques used by Hoogendoom [27] and Baxter [30,31]. 

### 2.9. Data Synthesis

As per the protocol, we undertook a narrative synthesis using the SwIM guidelines [32] (additional file 7) to guide reporting. This was in anticipation of heterogeneity in the variety of exposures, analysis methods and outcomes in the studies. Each study was assessed and associations between exposures and outcomes recorded as either “positive”, “negative” or “no association”. Studies were grouped by outcome and, within this, organised by exposure domain and tabulated to illustrate both the associations and assigned quality. The groupings for the outcome measure were undertaken by allocating the measure into either mental health, obesity or academic outcomes. The grouping for exposure measures was an inductive process involving an interpretation of the way child development had been understood and measured in each included paper, and then classifying and allocating these into a particular domain of child development; namely, a socio-emotional domain, cognitive domain, language and communication domain, physical domain or multiple domains. This was a subjective process because, as indicated in the introduction, there is no existing framework for understanding child development and characterising measures of child development. 

An overall rating on the strength of the evidence for each grouping (studies allocated within each domain of child development for each outcome: weight, mental health, academic) was derived as described in the quality assessment section. The results for factors which mediate or moderate associations between child development and subsequent health in adolescence (review question 2) was synthesised in relation to the conceptual diagram (additional file 2) of the relationship (produced with stakeholders at the outset of the review). Factors were classed as either mediators (those that explain associations) or moderators (those that alter the strength of associations) and assigned to a pathway (grouping of factors): family stress, knowledge/literacy, social/cognitive, material living and parent health behaviours. The overall ratings on the strength of the evidence for each domain and outcome, and stakeholder discussions, were used to inform a final diagram of the relationship between child development and adolescent health. 

## 3. Results

### 3.1. Literature Results

Following the screening of 10,657 retrieved citations, 34 articles were included in the review. See Figure 1 for PRISMA diagram illustrating the study selection process. Fifty-two studies were excluded on full text review; the list of studies excluded, with the reason, is available in additional file 4.

### 3.2. Study Design and Setting

Of the 34 included studies there were 32 prospective longitudinal studies [33,34,35,36,37,38,39,40,41,42,43,44,45,46,47,48,49,50,51,52,53,54,55,56,57,58,59,60,61,62,63,64], one retrospective longitudinal study [65] and one meta-analysis [66]. Detailed descriptions of the included studies are available in additional file 8. Of the 34 studies, 14 were set in the United States [35,36,39,43,45,47,49,50,51,56,57,58,59,64], seven in Canada [38,46,52,53,54,55,63], five in Australia [34,41,44,48,62], three in the UK [40,42,61], three in The Netherlands [33,37,65], one in Denmark [60] and one in which the countries included in the analysis were not explicitly stated [66].

### 3.3. Sample Size and Participant Characteristcs

The total number of children in included studies in the review was 69,152 (48% female, in those where sex was reported). Participants were recruited from pre-birth (through mother’s pregnancy) to age 12 years, with the majority recruited between the ages of 4–6 years, at pre-school or kindergarten. Across the studies recruitment took place between 1986 and 2009. The majority of the children were enrolled in existing longitudinal studies, were mainly Caucasian and from a mix of socioeconomic backgrounds. Six studies focused on socioeconomic disadvantage; three were of children from socio-economically disadvantaged families recruited from child care centres [50] or Head Start programmes (early years services to support low-income children and families in the US) [39,58], two studies oversampled for greater socioeconomic risk [51,61] and one oversampled for non-marital status [47]. A further two studies had children from majority low income [42] and low to middle income families [49]. There were three studies in which children from socioeconomic disadvantage were less well represented [34,38,52]. Children were assessed either in their own homes, pre-school or school apart from in two studies where lab-based assessments were made [46,51] and two where routinely collected healthcare data was used [61,65]. 

### 3.4. Studies Identified across Different Domains of Child Development (Exposures) and Adolescent Outcomes

Studies were found that focused on all domains of child development, namely: socio-emotional development, cognitive development, language and communication, and physical development. Table 2 illustrates the number of studies within each domain and the related adolescent outcome measure(s). Table 3 provides a summary of the main study characteristics and describes the exposures by domain of child development, outcomes and how they were measured. The main domain of child development studied in included papers was socio-emotional development with 24 studies [33,34,35,37,38,39,42,44,45,46,47,48,50,53,54,56,57,58,59,60,62,64,65,66]. Exposures included behaviours such as internalizing and externalizing behaviours, social competence, emotion knowledge, emotional wellbeing, emotional reactivity and peer relations. Exposures within the socio-emotional domain were generally measured using the relevant sections of standardized instruments such as the Child Behaviour Checklist (CBCL), the Social Behaviour Questionnaire (SBQ) or the Strengths and Difficulties Questionnaire (SDQ), with a mixture of child report, teacher report and parent report across the studies. 

Four studies [40,51,52,63] had an aspect of cognition as the exposure of interest, namely: mathematics skills, executive control, foundational cognitive ability, verbal ability/literacy and Intelligence Quotient (IQ). Executive control refers to a set of cognitive processes necessary for cognitive control of behaviour and was measured by observing tasks. Verbal ability was measured using literacy tests, mathematics skills by number knowledge tests or standardized assessments relating to the relevant country’s curriculum, and foundational cognitive ability and IQ by standardized instruments. 

Two studies [36,43] had language and communication as the main exposures and a further study [52] included language as one of multiple exposures. Exposures included receptive and expressive vocabulary. These were measured using the relevant sections of standardised assessments such as the Peabody Picture Vocabulary Test.

Two studies [49,55] incorporated exposures in the physical domain of child development. Exposures included fundamental movement skills (balance, agility, hand-eye co-ordination) and participation in structured and unstructured physical activity. These were measured by either parent report or assessment of skills by assessors in the child’s home. 

Two studies [41,61] measured across all domains of child development and education. One study assessed the component parts of teacher-rated school readiness in relation to the country’s early development instrument and one focused on child development in all domains in a health visitor check as a composite measure. In the main, studies analysed the effect of the exposure at a certain time point on an outcome at one later time point. However, two studies repeated measures at subsequent ages to assign children to a trajectory for the exposure of interest [38,58] and four studies repeated measures to study trends over time [43,48,57,64].

**Table 2 ijerph-18-11613-t002:** Studies by child development domain and adolescent outcome.

Number of Studies by Exposure Domain		Outcome Measures		
		Primary		Secondary
Domain:	Total studies	Mental Health	Weight	Academic
Socio-emotional	24 *	18	5	3
Cognitive	4 *	3	1	1
Communication and Language	2	2	1 ^	-
Physical	2	1	1	-
Composite/All domains measured	2 *	2	1	-
	34	26	9	4

* Includes one study which measured several outcomes. ^ From a study centrally coded to a different domain due to multiple exposures studied.

**Table 3 ijerph-18-11613-t003:** Summary of study characteristics.

Author (Citation)	Study Design	Country	Participants (% Females)	Exposure (Development Characteristic) and Age	Exposure Measurement Instrument	Outcome and Age	Outcome Measurement Instrument
Ashford et al. [33]	Longitudinal	Holland	294 (49.2)	Behaviour internalising and externalising—age 4	Child Behaviour Checklist (CBCL)—parent and teacher rated.	Internalising behaviours— age 11	CBCL—parent and teacher report.
Berthelsen et al. [34]	Longitudinal	Australia	4819 (49.1)	Child behaviour at age 4–5 and early ecological risk factors SEP, MMH, parenting anger, parenting warmth, parenting consistency	Child behaviour risk index measured as the sum of scores: sleep (emotional and dysregulation (both parent report) and inattention/ hyperactivity symptoms (mother rated).	Executive Function (age 14–15)	A composite score from three computerised tasks for assessing cognition (visual attention, visual working memory and spatial problem solving).
Bornstein et al. [35]	Longitudinal	US east coast	118 (42.0)	Social competence at age 4	Social competence as a construct, of: the peer acceptance subscale of the Pictorial Scale of Perceived Competence and Social Acceptance Preschool Form, the Friendship Interview, and the socialization domain of the Vineland Adaptive Behavior Scales (VABS).	Internalising and externalising behaviours at age 10 and 14	At age 10 years—the CBCL and Teacher Report Form At age 14 years—the CBCL and Youth Self-Report
Bornstein et al. [36]	Longitudinal	US east coast	Two studies Study 2 extracted— 139 (39.6)	Language—communication skills—at age 4	Two verbal subtests of the Wechsler Preschool and Primary Scale of Intelligence—Revised and the VABS.	Internalising and externalising behaviours at age 10 and 14	At age 10 years—the CBCL and Teacher Report Form At age 14 years—the CBCL and Youth Self-Report
Derks et al. [37]	Cohort	The Netherlands	One study of three extracted: Generation R study, 3794 (50.4)	Aggressive behaviour— at ages 5–7, 10 and 14	CBCL—mother rated	BMI and body composition (fat mass and fat free mass)—at ages 6 and 10	BMI—the Dutch national reference in the Growth Analyser program. FM and FFM—dual-energy X-ray absorptiometry scanner
Duchesne et al. [38]	Longitudinal	Canada	2000 (49.9)	Behaviour—hyperactivity, inattention, aggressiveness and prosociality—age 6Maternal warmth and maternal control also studied	Social Behaviour Questionnaire (SBQ)—teacher rated	Trajectory of anxiety at age 11–12	Rated annually from kindergarten to Grade 6 using the Anxiety Scale from the SBQ—teacher report Children put into trajectory of anxiety
Fine et al. [39]	Longitudinal	US	154 (50.0)	Emotional knowledge, internalising and externalising behaviours age 7	Emotion knowledge— composite score from two tasks: (Emotional labelling & Emotion situation knowledge) Internalising and externalising behaviours—CBCL (teacher report)	Internalising behaviours age 11	Child self-report aggregate of the following measures: Depression—Children’s Depression Inventory (CDI) Anxiety—The State-Trait Anxiety Inventory Loneliness. The Loneliness Scale Negative emotions—Differential emotions scale
Glaser et al. [40]	Longitudinal	UK	5250 (50.7)	IQ age 8	Wechsler Intelligence Scale for Children	Depression symptoms—age 11, 13, 14 and 17	Self-reported depressive symptoms were measured with the 13-item Short Mood and Feelings Questionnaire (SMFQ) Moderator: Pubertal stage at 11, 13 and 14 years was measured using a five-point rating scale
Gregory et al. [41]	Longitudinal	Australia	3906 (49)	School readiness across 5 domains (physical, social, emotional, language and cognitive, communication and general knowledge)—age 5	Australian version of the Early Development Instrument—teacher rated. Children scored as vulnerable, at risk or on-track	Age 11: four aspects of student wellbeing (life satisfaction, optimism, sadness and worries)	Middle Years Development Instrument—child self-report
Hay et al. [42]	Longitudinal	UK	134 (53)	Co-operation (one form of prosocial behaviour) at age 4	Tester’s rating of cooperativeness during the cognitive test (Tester’s Rating of Children’s Behaviour) and an observational measure of cooperation with the mother during the Etch-A-Sketch task	Internalising and externalising behaviour problems—at age 11	SDQ and CAPA (Child and Adolescent Psychiatric Assessment)
Hooper et al. [43]	Longitudinal	US	74 (52.7)	Language—receptive and expressive language, receptive vocab and working memory—age 5 and 7–8 (kindergarten and second grade)	Receptive and expressive language -The Clinical Evaluation of Language Fundamentals. Receptive vocab (Peabody test) and Working memory (Competing Language Processing Task)	Behaviour problems— externalising problems (conduct and hyperactivity)—kindergarten, first, second, and third grade	Teachers completed assessments of the children’s behaviour using a standardized scale of behaviour—Conner’s’ Teacher Rating Scale-Revised
Howard et al. [44]	Cohort	Australia	4983 (49)	Self-regulation—age 4–5 and 6–7	Self-regulation problems were indexed by combining parent-, teacher-, and interviewer-report ratings of children’s self-regulatory behaviours	Academic and weight, mental health, substance use, crime, self-harm and suicidal ideation—age 15	Academic achievement—children’s total scores on the Year 9 National Assessment Program—Literacy and Numeracy Mental health problems were measured in a private face-to-face interview with the parent/carer who knew the adolescent best Overweight and obesity—BMI
Howes et al. [45]	Longitudinal	US	307 (49.5)	Preschool social—emotional climate, Peer play, Behaviour problems, Teacher-child relationship quality—age 4	Preschool social—emotional climate—average of children’s scores on measures in class. Peer play–peer play scale Behaviour problems—classroom behaviour inventory (CBI) Teacher-child relationship quality—The Pianta Student Teacher Relationship Scale	Social competence—Behaviour with peers at age 8	Teacher reports using the Cassidy and Asher Teacher Assessment of Social behaviour Questionnaire
Jaspers et al. [65]	Longitudinal (retrospective)	Holland	2139 (50.9)	Behavioural features at age 4—”sleeping, eating, and enuresis problems” and “emotional and behaviour problems”	Assessed by Preventative Child Healthcare professionals.	Behavioural and emotional problems at age 10 to 12	CBCL—parent completed
Lecompte et al. [46]	Longitudinal	Canada	68 (48.5)	Emotional wellbeing—Child-parent attachment at age 3–4	Lab based separation reunion procedure	Anxiety and depressive symptoms and self-esteem (age 11–12)	Dominic Interactive Questionnaire- computerised self-report measure of common mental health disorders in childhood. Self-esteem–self-perception profile for children—self-report
Lee et al. [47]	Longitudinal	US	762 (46.3)	Behaviour internalising and externalising— age 5	CBCL—primary caregiver completed	Behaviour internalising and externalising—age 9	CBCL—primary caregiver completed
Louise at al. [48]	Longitudinal	Western Australia	2900 (not stated)	Behaviour— aggressive—age 5, 8, 10 and 14	CBCL, youth self-report at age 14 and teacher report at age 10 and 14	Weight at age 5, 8,10 and 14	Weight—Wedderburn digital chair scale Height was measured using a Holtain Stadiometer. BMI was calculated as weight (kg)/height^2^ (m^2^)
McKenzie et al. [49]	Longitudinal	USA	207 (49.7)	Fundamental movement skills—Balance, agility, eye-hand coordination—age 4,5 and 6	Movement skill tests in the child’s home	Physical Activity—age 12	Trained assessors administered the 7-day Physical Activity Recall (PAR) in the child’s home on two occasions, approximately 6 months apart
Meagher et al. [50]	Longitudinal	USA	56 (55.4)	Socio-emotional behaviours observed in pre-school—age 4	Externalising and internalising symptoms from the CBCL—teacher report Observed negative effect by research assistants	Depression symptoms— age 8	Child depression inventory—self-report
Nelson et al. [51]	Longitudinal	US	280 (47.9)	Executive control and Foundational Cognitive Abilities at age 5	EC–9-tasks administered to each child during individual sessions in the laboratory (working memory, inhibitory control, and flexible shifting) FCA—via the Woodcock-Johnson-III Brief Intellectual Assessment	Depression and Anxiety symptoms— Age 9–10.	Child Depression Inventory—child self-report Anxiety symptoms—Revised Child Manifest Anxiety Scale—child self-report Externalising symptoms—parents completed the ODD and ADHD-Hyperactivity subscales of the Conners 3rd Edition Parent Ratings Scale
Pedersen et al. [53]	Longitudinal	Canada	551 (45.4)	Behaviour—anxiety/ social withdrawal and disruptive behaviour—age 6	Social Behaviour Questionnaire (SBQ)—mother and teacher rated	Peer rejection & Friendedness (at age 8 to 11) Depressive symptoms Loneliness Delinquency—at age 13	Peer rejection–peer nominations. Friendedness—Children were also asked to nominate up to four best friends Depressive symptoms—CDI—child report Loneliness-self-report measure developed by Asher et al. 1984 Delinquency—Self-Reported Delinquency Questionnaire (SRDQ)
Piche et al. [54]	Longitudinal	Canada	966 (47.0)	Self-regulatory skills: classroom engagement and behavioural regulation (emotional distress, physical aggression, impulsivity)—age 6	Classroom engagement (teacher rated) and Behavioural regulation using the SBQ (teacher rated)	Child Sports Participation and BMI— Age 10	Parents reported on their child’s weekly involvement in structured sports outside of school during the past school year BMI was derived from direct height and weight measures made by trained, independent examiners
Piche et al. [55]	Longitudinal	Canada	1516 (51.9)	Participation in structured and unstructured physical activity—age 7	Parents reported on their children’s participation in structured and unstructured physical activity	Age 8 Depressive symptoms	Depression symptoms assessed through the Social Behaviour Questionnaire
Rudasill et al. [56]	Longitudinal	USA	1156 (48.8)	Child temperament (negative emotionality at age 4½ and emotional reactivity at age 7–12) (Student-teacher relationship -teacher perception and child perception tested as mediators)	Negative emotionality: Mothers completed eight subscales from the Children’s Behaviour Questionnaire Emotional reactivity: Children’s emotional responses to events and environmental stimuli were rated by mothers using a measure designed for use in the NICHD SECCYD	Depressive symptoms in sixth grade (age 11–12)	Mother report of their children’s depressive symptoms was measured in 6th grade with the Diagnostic and Statistical Manual of Mental Disorders oriented Affective Problems subscale of the Child Behaviour Checklist
Rudolph et al. [57]	Longitudinal	USA	433 (55.0)	Peer Victimization (static and dynamic) (Age 7–12, 2nd to 5th grade)	Children and teachers completed a revised version of the Social Experiences Questionnaire to assess children’s exposure to peer victimization.	Depression symptoms and Aggressive behaviour—Age 11–12 (5th grade)	Depression symptoms—Short Mood and Feelings Questionnaire (Child report) Aggressive behaviour—Children’s Social Behaviour Scale (teacher report)
Sandstrom et al. [66]	Meta-analysis	Any	8836 (51.5)	The mean age at the first BI assessment was 3.61 years	BI: defined as shyness, fear, and avoidance when faced with new stimuli	The mean age at the anxiety assessment was 10.39 years	Anxiety and specific anxiety types searched
Sasser et al. [58]	Longitudinal	USA	356 (54.0)	Intervention targeting social-emotional functioning and language-emergent literacy skills in the first year of pre-school. Executive function measured before and after preschool and each year to third grade (age 8)	Executive function assessment by trained examiners. Children assigned to either low, moderate or high executive function trajectory	Third grade academic outcomes	Reading fluency, language-arts and mathematics (all teacher rated), children self-evaluation of reading ability
Shapero et al. [59]	Longitudinal	USA	958 (48.0)	Emotional—emotional reactivity at age 8. (Household income and household chaos also studied.Household Chaosand Household income also studied.)	Emotional reactivity—mother report—10-item questionnaire about their perceptions of how their child expresses emotions in response to events	Emotional and behavioural problems— age 15	Adolescent Emotional and BehaviouralProblems—Youth Self-Report.
Slemming et al. [60]	Longitudinal	Denmark	1336 (49.0)	Behaviour: anxious–fearful, hyperactive–distractible, and hostile– aggressive—age 3–4	Preschool behaviour questionnaire (PBQ)—parent report	Internalising problems— age 10–12	Emotional difficulties were measured at age 10–12 years with the parent-administered strength and difficulties questionnaire (SDQ)
Straatmannet al. [61]	Longitudinal	UK	10262 (not stated)	Five central domains of a health check in England: (1) personal, social and emotional development, (2) communication and language, (3) physical health, (4) learning and cognitive development and (5) physical development and self-care)—at age 3	Health visitor assessment at routine health check	Language, weight, socioemotional behaviour— age 11	Language—British Ability Scale Second Edition (BAS II) Verbal Similarities test Weight was derived from the body mass index (BMI), using the age and sex- International Obesity Task Force cut-offs Socio-emotional behaviour—SDQ—mother report
Sutin et al. [62]	Longitudinal	Australia	4153 (71.6)	Temperament— sociability, persistence, negative reactivity. Age 4–5	Parents completed a 12-item measure of temperament based on the Childhood Temperament Questionnaire	Weight and weight attitudes and behaviour—age 14–15	Weight—BMI and waist circumference at all ages Weight attitudes and behaviour. At ages 14–15 years, study children self-reported on several aspects of their attitudes and behaviours.
Weeks et al. [63]	Longitudinal	Canada	4405 (50.0)	Verbal ability (age 4–5) and Math skills— age 7–11	Verbal Ability: Peabody Picture Vocabulary Test-Revised (PPVT-R) Math skills—Mathematics Computation Test (MCT)	Internalising symptoms of anxiety and depression—age 12–13 and 14–15	Questionnaire that included 7 items from the Ontario Child Health Study (OCHS-R), assessing symptoms of anxiety and depression—self-report.
Yan et al. [64]	Longitudinal	USA	695 (49.1)	Emotional Wellbeing—child parent relationship—Age 6	Both fathers and mothers rated their relationships (conflict and closeness) with the child at Grade 1, 3, 4 and 5—Child-Parent Relationship Scale	Loneliness at grades 1, 3 and 5 (age 10–11)	Loneliness and Social Dissatisfaction Questionnaire—child self-report

### 3.5. Quality Assessment

Thirty-three of the 34 included studies were assessed using the methodological assessment tool for observational studies, available in additional file 6. One study, a meta-analysis, was assessed using AMSTAR (A MeaSurement Tool to Assess systematic Reviews). Results of the quality assessment process for all included studies is available in additional file 9. Ten were rated as low, 16 moderate and eight high in methodological quality. High implies a low risk of bias, moderate implies a moderate risk of bias and low quality implies a high risk of bias. 

As outlined in quality assessment section of the methods, confidence in cumulative evidence was assessed within each grouping of papers, grouped by outcome and domain. This is referred to throughout the synthesis of the findings.

### 3.6. Narrative Synthesis

There was a range of exposures and outcomes reported across the included literature. Studies were organised by outcomes and grouped as follows: “Mental health related symptoms”—this incorporated: internalising symptoms (general, depression, anxiety, loneliness and self-esteem), externalising (general and ‘delinquency’), socio-emotional behaviour problems, social competence, wellbeing, self-harm and suicidal ideation.“Weight, diet and physical activity”—this incorporated: BMI, overweight/obese, sports participation, unhealthy weight attitudes, and healthy dietary habits.For secondary outcomes, the group included executive function and outcomes from academic tests.

Within these above groupings, studies were subsequently organised by exposure and by each domain of child development as follows: Domain: Social and emotional development. This was further subdivided to aid analysis, as follows:
○Internalising—internally focused behaviour such as inhibition and withdrawal. ○Externalising—externally focused behaviour such as aggression, attention problems, hyperactivity and “delinquent” behaviour.○Emotional—internal factors such as social competence, emotion knowledge, pro-social, co-operative and self-regulation skills. External factors such as peer relations, parent-child relationships, teacher-child relationships, socio-emotional climate of school/pre-school setting.○Temperament—negative emotionality, emotional reactivity and persistence.Domain: Language and communication. This comprised the ability to listen, understand and speak. Exposures included: receptive and expressive vocabulary. Receptive relates to understanding of words and expressive relates to the ability to use words for expression.Domain: Cognitive development. This comprised mathematics skills, executive control, foundational cognitive ability, verbal ability/literacy and Intelligence Quotient (IQ)Domain: Physical development. This involved fundamental movement skills (balance, agility, hand-eye co-ordination) and participation in structured and unstructured physical activityMultiple domains.

A summary of the evidence on associations between exposures (domains of child development) and outcomes is presented in Figure 2. Each annotation does not always represent a study in its entirety as many studies analysed multiple exposures and outcomes. 

### 3.7. Primary Outcomes

#### 3.7.1. Mental Health 

##### Summary of Associations between Child Development and Mental Health

Positive development on starting school is associated with subsequent positive mental health. There is stronger evidence for associations between the socio-emotional domain of child development and later mental health, weaker evidence for the cognitive domain, inconsistent evidence for language and communication and limited evidence for physical development.

##### Summary of Associations between Socio-Emotional Development and Mental Health

Eighteen studies analysed associations between a socio-emotional exposure of child development and later mental health [33,35,38,39,42,44,45,46,47,50,53,56,57,59,60,64,65,66]. All associations highlighted that positive socio-emotional development is good for subsequent mental health, apart from five studies where no associations were found for some exposures and outcomes studied [39,42,50,60,65]. The evidence is stronger for exposures of externalising behaviour and emotional wellbeing at school entry, weaker for exposures of internalising behaviour and limited for exposures relating to temperament. 

##### Exposure of Internalising Behaviours at School Entry and Subsequent Mental Health

Eight studies analysed the relationship between early internalising behaviour and later mental health [33,39,47,50,53,60,65,66], highlighting weaker evidence for positive associations with internalising outcomes and limited evidence for positive associations with externalising outcomes. Of these, six studies analysed the association between early internalising and later internalising behaviours, with two studies of moderate quality showing positive associations [33,47], one high quality study where no association was found [39], and one low quality study, of 56 children, where no association was found with depression symptoms [50]. Specifically, anxious-fearful behaviour is associated with later emotional difficulties as reported by parents [60] in a study of moderate quality. Behavioural inhibition is associated with anxiety but this evidence was from a lower quality review [66]. Evidence on the relationship between early internalising and later externalising behaviours was scant; only limited evidence was provided, with two studies not studying that relationship specifically [47,53] and one study where no association was found between early emotional and behaviour problems and later externalising [65].

##### Exposure of Externalising Behaviours at School Entry and Subsequent Mental Health 

Nine studies were found on the relationship between externalising behaviours and later mental health [33,38,39,45,47,50,53,60,65], highlighting stronger evidence for positive associations with both internalising and externalising outcomes. Of these, seven studies analysed the associations between early externalising and later internalising, with six studies showing positive associations and one study where no association was found. There was evidence of positive associations between general externalising behaviour problems [33,39] and later internalising symptoms, and specifically aggression [60] was associated with later internalising symptoms. However, although two studies showed no association between hyperactive behaviour [60] or inattention [65] and later general internalising symptoms, one high quality study of 2000 children did evidence an association between these behaviours and later anxiety symptoms [38]. One study, of lower quality, evidenced that disruptiveness was associated with later depression symptoms and loneliness [53]. 

Similar to internalising symptoms, whereby the continuity of association was found for early and later symptoms, the same is true for externalising symptoms, whereby early problems are associated with externalising at a later age. However, the evidence is stronger with two studies of moderate quality evidencing associations between general externalising [47], inattention and behaviour problems [65] and later general externalising symptoms, with a further study evidencing a relationship between poorer social competence with peers in mid-childhood and earlier behaviour problems [45]. Specifically, disruptiveness was associated with delinquency in one low quality study [53].

##### Exposures of Emotional Wellbeing at School Entry and Subsequent Mental Health

Nine studies were found on the associations between a child’s emotional wellbeing and later mental health [35,38,39,42,44,45,46,57,64] with stronger evidence found for the association with internalising outcomes and weaker evidence for externalising outcomes. A child’s emotional wellbeing, in terms of social competence, emotional knowledge (the ability to identify and label emotions), self-regulation and prosociality (behaviour intended to benefit others), appears beneficial to later health in adolescence. Negative associations were found between early social competence and internalising and externalising problems [35]. In two high quality studies, associations were found for emotional knowledge [39] and prosocial skills [38] and later anxiety, with increasing emotional knowledge and prosocial skills both associated with less anxiety symptoms. A child’s ability to co-operate, a particular prosocial skill, highlighted mixed results in one study [42], with increasing co-operation associated with less externalising but no association found with internalising problems. Self-regulation problems, in terms of ability to control behaviours, attention, thinking, social interaction and emotions, were subsequently associated, in adolescence, with an increase in the risk of self-harm ideation and behaviour, suicidal ideation, school truancy, mental health problems, smoking and alcohol use, and violent and property crime [44]. When self-regulation problems reduced, from age 4–5 to 6–7, the association between these adolescent outcomes and earlier self-regulation problems was no longer found [44].

In relation to the child’s emotional wellbeing in the context of relationships or setting specific (external factors), studies were found on: mother–child attachment, relationship with parents, teachers, and peers (victimisation), and the socio-emotional climate in a pre-school setting, and all proved important for positive mental health in adolescence. A small study of 68 children, rated low quality, evidenced that disorganised maternal attachment at pre-school age was associated with greater depression and anxiety symptoms and lower self-esteem in early adolescence [46]. A positive relationship with parents, in terms of closeness, was associated with less loneliness, particularly for father–daughter relationships [64]. In one low quality study, a good quality relationship with teachers and a positive socio-emotional climate in a pre-school setting were both associated with improved social competence in mid-childhood [45]. With regard to relations with peers, one study evidenced that early and increasing peer victimisation was associated with depression symptoms and aggression [57].

##### Exposures of Temperament at School Entry and Subsequent Mental Health

Two studies were found on the association between temperament and later mental health highlighting limited evidence of a negative association, with higher levels of certain traits associated with worse outcomes. These studies investigated child temperament, in terms of negative emotionality and emotional reactivity (the former refers to the propensity to react with negative emotions and the latter relates to the intensity of emotion) [56,59] and both were of moderate quality. One showed an association between negative emotionality, emotional reactivity and depression symptoms [56] and one between emotional reactivity and internalising and externalising symptoms [59]. 

###### Summary of Associations between Language and Communication, Cognitive Development, Physical Development, and Multiple Domains and Mental Health

Eight studies analysed the associations between exposures relating to either language and communication, cognitive development, physical development or multiple domains of child development and later mental health. All associations highlighted that positive development across all of the domains of child development are good for subsequent mental health. There was weaker evidence for the effect of cognitive skills and the positive effect of cognitive development appears to alter with age. The evidence for associations between language and communication and later mental health outcomes was inconsistent in relation to internalising and externalising outcomes. There was limited evidence for both physical development and measures incorporating multiple domains.

##### Exposures within the Language and Communication Domain and Subsequent Mental Health

The results for the effect of language and communication skills on later mental health symptoms was inconsistent with two studies investigating these associations [36,43]. One study of 129 children evidenced that language skills at pre-school age predict internalising but not externalising behaviour problems in adolescence. Conversely, one low quality study of 74 children did find an association between good language skills (receptive and expressive language) and less externalising problems, namely, conduct problems but not hyperactivity.

##### Exposures within the Cognitive Domain and Subsequent Mental Health

Three studies analysed the effect of cognitive skills on later mental health symptoms, [40,51,63] with weaker evidence found. One study found that deficits in executive control predicted depression and anxiety symptoms and clinical level of depression [51]. The same study showed that foundational cognitive ability did not predict these outcomes. One high quality study showed an association between cognition, measured as IQ, and depression symptoms with an increased IQ in early childhood associated with less depression symptoms at age 11 [40]. However, by age 13–14 the association reversed. The loss of protective effect of cognition was also found in relation to the effect of cognitive skill (measured as mathematics skills and verbal ability) on internalising symptoms, whereby a protective effect seen at age 12-13 was reversed or had no associated effect at age 14–15 [63].

##### Exposures within Physical Development Domain and Subsequent Mental Health

There was limited evidence for the effect of physical development on later mental health related symptoms. One lower quality study, in which time between exposure and outcome was one year, found that structured physical activity was associated with less depression symptoms in boys, whereas unstructured physical activity was associated with more depression in girls [55]. 

##### Exposures Incorporating Multiple Domains and Subsequent Mental Health

Two studies provided evidence across multiple domains [41,61]. One study evidenced that all components of school readiness (as part of a model of early years data), measured by UK health visitors before starting school, predicted socio-emotional behaviour problems in early adolescence [61]. An Australian study which investigated the relationships between all domains of school readiness and wellbeing at the end of primary school found that all domains were negatively associated with internalising symptoms, whereas only physical and socio-emotional development were positively associated with overall wellbeing [41].

#### 3.7.2. Weight, Diet and Physical Activity Outcomes 

##### Summary of Associations between Child Development and Weight

Positive development on starting school is associated with subsequent healthy weight related outcomes. There is stronger evidence for the socio-emotional domain of child development, and limited evidence for language and communication, cognitive and physical domains of child development.

##### Summary of Associations between Socio-Emotional Development and Weight 

Five studies analysed the associations between a socio-emotional measure of child development and later weight diet or physical activity outcomes [37,44,48,54,62]. All associations highlighted that positive socio-emotional development is good for subsequent weight-related outcomes, apart from one study where mixed associations were found for exposures of certain temperamental traits and later weight related outcomes. The evidence is stronger for exposures within the emotional wellbeing domain, specifically self-regulation skills, with weaker evidence found for exposures of externalising behaviour and no evidence found for internalising behaviour. 

##### Exposures of Externalising, Emotional Wellbeing and Temperament and Subsequent Weight

In relation to externalising, specifically aggressive behaviour, one higher quality study found a positive association with higher BMI [37] and one of moderate quality found an association with higher rate of change in BMI but in girls only [48]. In relation to self-regulation, one higher quality study [44] evidenced that early problems in self-regulation (ability to control attention, behaviour and emotion at age 4–5) were associated with being overweight or obese in adolescence but that a change in self-regulation (less problems) at a later age (age 6–7) had no effect on the association. Another study highlighted that increasing self-regulation skills (measured as class room engagement) were associated with lower BMI and increased sports participation [54]. Additionally, this study evidenced that emotional distress (a measure of self-regulation) was associated with less sports participation. In relation to temperament, one higher quality study [62] looked at the associations between the traits of persistence sociability and negative reactivity, and later BMI and weight attitudes and behaviours, and found that persistence decreased the risk of obesity and overweight, sociability increased the risk of overweight but not obesity, and negative reactivity was not associated with either. In relation to weight attitudes and behaviours, all three traits were associated with restrained eating habits in adolescence, with lower persistence and higher negative reactivity or sociability associated with restrained eating and use of unhealthy weight management strategies. 

##### Summary of Associations between Domains of: Language and Communication, Cognitive, Physical Development, Multiple Domains and Weight 

There was limited evidence on associations between the domains of language and communication, cognitive skills, and physical development, and later weight-related outcomes. One study, which looked at evidence on a range of school readiness skills and later wellbeing measures, evidenced that receptive vocabulary was associated with healthier dietary habits [52], with increasing receptive vocabulary predicting reduced sweet snack intake and increased dairy intake. The same study [52] evidenced that increasing mathematics skills predicted increasing involvement in physical activity, providing limited evidence for an association between cognitive skills and later weight-related outcomes. There was limited evidence on the association between physical development and weight-related outcomes, with one lower quality study finding no association between fundamental movement skills and later involvement in physical activity [49]. One study evidenced that all components of school readiness (as part of a model of early years data), as measured by UK health visitors before starting school, predicted overweight and obesity in early adolescence [61].

### 3.8. Secondary Outcomes

#### 3.8.1. Academic Tests and Executive Function

##### Summary of Associations between Child Development and Academic Outcomes

Four studies analysed associations between a domain of child development and later academic outcomes: three in relation to socio-emotional development and one in relation to cognitive development. All associations highlighted that positive development is good for subsequent academic outcomes. The evidence is stronger for exposures within the socio-emotional domain, specifically self-regulation skills and less behaviour problems, with weaker evidence found for exposures within the cognitive domain of child development. There were no studies found looking at the association between language and communication or physical development and academic outcomes.

##### Exposures within the Socio-Emotional and Cognition Domains and Subsequent Academic Outcomes

There were two studies found on associations between socio-emotional development and the secondary academic outcomes, both of higher quality [34,44]. One studied the effect of behaviour risk (a composite of poor sleep, emotions and inattention) on adolescent executive function and found that poorer behaviour is associated with lower executive function [34]. Another study highlighted that self-regulation problems are associated with reduced scores on numeracy and literacy tests in adolescence [44]. An additional study [58] investigated the effect of an intervention targeting socio-emotional functioning and language emergent literacy skills in pre-school, through comparing the impact of executive function trajectories on academic test results of children in the intervention group compared to those who were not. This study showed that socio-emotional and language programmes improved executive function and academic outcomes for children with the lowest executive function trajectory. There was limited evidence on associations between early cognitive skills and later academic outcomes, with one study showing a positive association between kindergarten mathematics skills and later academic outcomes [52]. 

### 3.9. Factors Affecting Relationships

Summary of findings on factors affecting associations (mediation and moderation)

Limited evidence was found on factors affecting associations. Some evidence, however, was found on factors affecting associations between socio-emotional development and subsequent mental health and academic outcomes. The factors are discussed in relation to the pathways identified in the initial conceptual model devised with stakeholders. Factors were found in relation to family stress, knowledge/literacy and social/cognitive pathways. No factors were found that pertained to the material living or parent health behaviour pathways. All of the findings within this section fall into the category of limited evidence as all of the factors described were found in single studies only.

#### 3.9.1. Mediators

Six studies included data on mediating variables: five related to studies focusing on mental health outcomes [35,36,46,53,56] and one relating to academic outcomes [34]. None of the studies that focused on weight as an outcome included data on mediation. Factors mediating associations between socio-emotional development and mental health were self-esteem, type of internalising or externalising in mid-childhood, and relationships with teachers and friends. Factors mediating associations between socio-emotional development and academic outcomes were approaches to learning and attentional regulation.

#### 3.9.2. Moderators

Seven studies included data on variables to test for moderation effects on associations between exposure and outcome: five in relation to mental health [38,40,47,59,64], one in relation to weight [62] and one in relation to academic outcomes [58]. Factors moderating associations between socio-emotional development and mental health were household chaos and parenting. Household chaos had a negative effect, and aspects of parenting had a positive effect on the associations between the socio-emotional domain of child development and mental health outcomes. A factor found to moderate the association between socio-emotional development and academic outcomes was trajectory of executive function.

#### 3.9.3. Factors Pertaining to the Family Stress Pathway—Moderating Associations between Child Development and Mental Health

This pathway incorporates factors related to stress in the home, which can affect parenting ability, parenting style and consequently child health and development. Household chaos and aspects of parenting were identified as moderators of the relationship between the socio-emotional domain of child development and later mental health symptoms. Household chaos was found to disproportionately affect children with higher emotional reactivity resulting in greater internalising problems [59]. This effect was not found for household income; that is, level of emotional reactivity made no difference to the impact of income on adolescent emotional and behaviour problems. This implies that the impact of low income on adolescent mental health is pervasive and not amenable to individual interventions promoting self-regulation (in terms of emotional response to events) but that interventions of this type might support how children respond to household chaos. Three studies analysed the moderating role of aspects of parenting on the relationships between socio-emotional measures and later mental health related symptoms [38,47,64], with all finding positive effects of aspects of child/parent relationships on adolescent outcomes. Two studies found a protective effect of relationships with fathers on continuity of behaviour problems. One found a protective effect for fathers’ positive engagement on the continuity of earlier to later internalising and externalising behaviour problems, and for those in the greatest poverty, fathers’ positive engagement was associated with a reduction in the continuity of internalising problems from age 5 to 9 years [47]. The authors hypothesize that this is due to development of secure attachment and the development of emotional and behavioural regulation skills. Another looked at the moderating role of parent–child closeness on the continuity of loneliness from age 6–11 years and found that as parent–child closeness increased, loneliness reduced and this relationship was particularly strong for girls and their fathers [64]. Another study looked at the moderating role of maternal parenting practices, warmth and discipline, on the relationship between behavioural characteristics of; inattention, hyperactivity, aggressiveness and low prosociality and trajectory of anxiety in children between the ages of 6 and 12 years [38]. It found that a lack of maternal warmth increased the association between hyperactivity and anxiety. It also found that high level of maternal discipline (rules and controlling child’s behaviour) increased the probability of belonging to the high anxiety group.

#### 3.9.4. Factors Pertaining to the Knowledge/Literacy Pathway—Moderating and Mediating Associations between Child Development and Academic Outcomes

This pathway relates to the way knowledge and literacy can lead to behaviours that can be positive for wellbeing [67]. Two studies analysed factors within this category, both on the relationship between the socio-emotional domain and later academic outcomes. One highlighted a moderating role of executive function on the relationship between an intervention to improve socio-emotional and language emergent skills and later academic outcomes, and found that the effect of the intervention was higher in children with low executive function in the intervention group resulting in better academic outcomes compared to controls [58]. Another highlighted attentional regulation and approaches to learning as mediators, partially explaining the relationship between child behaviour (composite of sleep, emotional dysregulation and inattention/hyperactivity) and later executive functioning. 

#### 3.9.5. Factors Pertaining to the Social/Cognitive Pathway—Mediating Associations between Child Development and Mental Health 

This pathway relates to the influence of individual experiences, the actions of others and environmental factors that provide the social context for learning to influence health behaviours [68]. Five studies analysed mediators pertaining to this pathway and all in relation to mental health related symptoms: four in relation to the socio-emotional domain [35,46,53,56] and one from the language and communication domain [36]. In relation to the socio-emotional domain of child development and later mental health there is limited evidence that self-esteem, type of internalising or externalising in mid-childhood, and relationships with teachers and friends all play a role in explaining the relationship. One study evidenced the role of self-esteem, which partially mediated the relationship between emotional wellbeing (measured as parental attachment) and depression but not anxiety [46]. Two studies highlighted the role of relationships: one with peers [53] and one with teachers [56]. In relation to peers, peer rejection and number of friends in mid-childhood mediated the relationship between disruptiveness at age 6 and depression at age 13 years, with peer rejection also mediating the relationship between disruptiveness and loneliness. Relationships with teachers (closeness and conflict) also appear important, with one study analysing the effect of child–teacher relationships on the relationship between negative emotionality at age 4, emotional reactivity at age 7 and depression symptoms at age 11–12 years [56]. This study found that teacher-child conflict mediated the relationship between emotional reactivity and depression symptoms, with children higher in emotional reactivity having more depression symptoms, and this was partially explained by conflict with teachers (teacher reported).

#### 3.9.6. Factors Pertaining to Child Characteristics—Moderating Associations between Child Development and Weight and Mental Health

Sex moderated the association between socio-emotional domain of child development and weight outcomes, with worse outcomes for girls. Age moderated the association between the cognitive domain of child development and subsequent mental health, with a protective role of positive cognitive development on mental health in early adolescence reversing in mid-adolescence.

Two studies analysed the effect of age on the relationship between cognition and later depression symptoms, and found that age reversed the protective effect of cognition on early adolescent (age 11) mental health by age 13–15 years [40,63] but that this reversed again at age 17 for females [40]. This study also found that pubertal status mimicked the relationship by age but that this was stronger for females. The loss of the health protective effect of cognition may be due to exam pressures at certain time points or, for females, biological hormonal changes. Sex was the only factor studied in relation to weight. One study evidenced that girls with higher aggression cores throughout childhood had a higher rate of change in their BMI [48]. Another study highlighted that girls higher in sociability in early childhood had a greater fear of weight gain at age 14–15 years [62]. 

### 3.10. Conceptual Model/Diagram Development

A summary of findings and revised conceptual model was discussed with stakeholders and a final diagram of the relationships, as informed by this systematic review, was discussed and agreed—see Figure 3. Factors which were not found in the review but were deemed important by stakeholders are highlighted on the diagram. These included neighbourhood factors, such as community engagement and community environment, which stakeholders felt could create conditions conducive for optimal health and development. Political and system factors were also identified, such as short political cycles not giving policy sufficient time to embed and effect change, and regulators focusing narrowly on academic outcomes rather than broader social and emotional wellbeing, which dictates the focus of a school. The stakeholder group identified these factors as potential moderators of the relationships between child development and health.

## 4. Discussion

This review asked the questions: what are the associations between child development and adolescent health, and what factors explain or alter the associations. The review clearly shows that positive development on starting school is good for later health outcomes, but that the evidence is stronger for relationships between some domains of child development than others, with gaps in the evidence base across domains. In relation to mental health outcomes, there is stronger evidence for associations between socio-emotional development and later mental health, weaker evidence for associations with cognitive development, inconsistent evidence for language and communication, and limited evidence for physical development. In relation to adolescent weight, there is stronger evidence for associations with children’s socio-emotional development and limited evidence of relationships with language and communication, cognitive development or physical development. In relation to secondary (academic) outcomes, there is stronger evidence for associations with socio-emotional development, limited evidence for an association with cognitive development and no evidence found for an association with language and communication or physical development. In relation to what factors explain or moderate the associations, the evidence identified in this review is largely limited to factors shaping the relationship between socio-emotional development and mental health and academic outcomes with factors pertaining to the pathways of family stress, knowledge/literacy and social/cognitive. 

Our findings build upon the existing limited evidence that attendance at pre-school (a proxy for good child development) is associated with positive mental wellbeing and healthy weight in adolescence [12], and provides detail on which domains of child development are associated with these positive health outcomes. Supporting the existing literature, we found positive relationships between both cognitive and socio-emotional development (such as self-regulation and social competence) on mental health and academic outcomes in early adolescence [9,10,11], with stronger evidence found for socio-emotional development. Additionally, we identified evidence of a negative relationship between socio-emotional wellbeing and unhealthy weight. The review provided a test of our conceptual model and we found that the evidence base was lacking for some of the proposed pathways between child development and later health. 

Undertaking this review highlighted a complexity in classifying the rich and broad literature that exists on child development. In consequence, the review had to embed an inductive process of interpretation of how included studies had understood and measured child development. Using a classification system of four domains of child development to aid analysis of a very broad concept enabled us to categorise a multitude of measures of child development within this system. This complexity and the need for interpretation is perhaps unsurprising given that the field of child development spans the disciplines of psychology, sociology education, biology, genetics and public health. That a classification system for understanding child development does not exist is in itself a finding of our review. Our classification of four domains of child development adds to the literature, providing a framework for other researchers to use and to critique. 

Our findings show that conceptualising child development into domains of development matters because different aspects of development seem to have different impacts on later health outcomes. Understanding this can help to inform public health interventions in childhood. For example, in our review, we found that socio-emotional development when children start school has the most evidence for subsequent impact on adolescent health, in terms of mental health and healthy weight, and as such could be a focus for intervention. The findings in relation to mental health are to be expected, with much literature highlighting the continuity of early problems with socio-emotional functioning and later onset of mental health conditions [69]. The evidence is stronger for early externalising behaviours and their impact on both internalising and externalising behaviours in adolescence, and this is supported by wider literature [70,71]. A finding of this review is that there is more evidence that behaviours such as aggression and hyperactivity pose a risk to future mental health than anxious/withdrawn behaviour, particularly for externalising outcomes. This finding should be interpreted cautiously because it may be that early internalising behaviours, compared to externalising, are more likely to resolve by early adolescence [72], or it may be that internalising is harder to identify, whereas externalising behaviours are more obvious and easier for parents and teachers to report, which could lead to less associations being observed for internalising behaviours and consequently less associations found [39,70]. The finding that emotional wellbeing was more closely associated with later internalising may be because emotional stability promotes regulation and mood stability leading to less internalising [73], but other studies have found that emotional wellbeing (in terms of regulation skills) is associated quite strongly with both internalising and externalising, but particularly so for internalising after the early years [74]. 

The findings in relation to stronger evidence on the associations between socio-emotional development and weight add to a growing field of evidence exploring this relationship, with evidence of co-development and temporal associations in mid-childhood [75], evidence of obesity having a detrimental impact on socio-emotional behaviour [76], and evidence on associations between social competence and weight with social competence reducing the odds of later overweight [77]. From this review, emotional wellbeing and, in particular, self-regulation skills, appear to be important factors to study in this complex relationship between socio-emotional development and weight. However, other developmental pathways in the development of obesity, such as physical activity and cultural and social factors, are important to consider alongside self-regulation development [44]. 

More evidence is needed on how adolescent health outcomes are shaped by other domains of child development, particularly the impact of language and communication, and cognitive and physical development at school starting age. Evidence on these domains are important for engaging health and education sectors to work together because education and health services share a common goal for optimal developmental potential of children [78]. This evidence would help the development of a shared understanding and provide a platform from which to develop the context and settings that may work best for optimal health and development of children, regardless of their stage of development when starting school. Including executive function as an exposure and outcome in this review allowed for inclusion of any evidence on the bi-directional relationship between executive function and health [79]. The analysis of secondary outcomes of academic tests and executive function highlight the importance of socio-emotional development on these outcomes (health improves executive function). Conversely, the protective effect of cognitive skills (measured as executive control) on adolescent mental health highlights that executive function improves health. However, age appears to be an important factor in this latter relationship, with the protective effects of cognition on mental health being reversed or no associations found in mid-adolescence [40,63], and this warrants further research. 

Understanding the impact of domains of child development on later health has important policy implications in relation to reducing inequalities, and in relation to a policy extension beyond the first 1000 days. In relation to reducing inequalities, our review highlights a strong relationship between socio-emotional development and later health. Applying a public health lens to “child development” helps to understand exactly what it is that pre-school provision or early years centres may need to focus on if we are to improve adolescent health and wellbeing. Re-invigoration of early childhood programmes such as Sure Start, with a renewed focus on socio-emotional development, may be one area of policy improvement, particularly if we are to focus on their longer-term potential to reduce inequalities [80]. 

In addition, arguably a policy shift is required, which extends beyond the first 1000 days, to understand and support optimal development throughout childhood and into adolescence [81], to address the consequences of inequalities in child development as children age and because adolescence is a significant period of development and an important period in the life course [82]. If we are to maximise the opportunities conferred by education as a platform to improve public health and reduce health inequalities [83], policy that incorporates a life course approach to healthy development is needed, and this requires cross-sector collaboration. 

Fostering collaboration to inform policy on reducing child and adolescent health inequalities beyond the first 1000 days requires more research on how development and education translates into health throughout childhood, and on the effect of socio-economic circumstances on this relationship. Findings in relation to the factors that explain or alter associations between child development and subsequent health were limited in this review, with all findings pertaining to single studies. The most evidence was within the social-cognitive pathway with self-esteem, relations with peers and teachers all providing some explanation for the relationship between socio-emotional development and subsequent mental health outcomes, and this can inform interventions for optimal health and development through the primary school years. Surprisingly, in relation to the original conceptual diagram designed with stakeholders, there was no literature found on material living circumstances, parent health behaviours, community factors, and political and system factors. To some extent this was because the studies controlled for the effect of income, housing, parental education and parent health behaviours. However, this presents a problem because we need to know more about how these elements of socio-economic circumstance affect the associations under study in this review. For example, we know that children from more deprived backgrounds experience poorer health and development than their more affluent peers [7]. If we are to pragmatically intervene to improve the health and development trajectory of children in more deprived circumstances, and reduce the attainment and health gap, we need to understand exactly how poverty, household circumstances and home environments affect learning and the co-development of education and health. This requires the design of public health research that respects agency but more clearly theorizes children within their social and economic context [84], so as to encapsulate socio-political cultural and familial environments because, in many ways, this is what “defines” child development in practice, over and above genetic make-up. 

In addition to collaborating to produce more evidence on individual, home or school-level interventions to mitigate against poor development, interventions at system level are required to tackle prevention earlier so that children reach a good stage of development, and to reduce inequalities in development measures upon starting school. This requires evidence in relation to macro-determinants such as political and system factors, e.g., addressing poverty, the role of regulators in generating a target-driven culture that focuses attention on academic achievement, a political system not conducive to cross-departmental perspectives or action, and a system which stifles innovation and creativity. Larger studies, such as natural experiments or evaluations of existing policies, are needed that perhaps compare areas with different working systems or policies to identify any particular cultures or practices that are conducive to promoting positive trajectories for the health and development of children. The participatory element of this research identified some of these macro-determinants, identifying a gap between research and practice. This finding highlights the need to bring research and practice closer together [85] through listening to views and experiences of those working in service roles, at the system level and/or with children. It is hoped that this method can help to inform future research by highlighting what is evidenced in the literature, but also bringing to light views from lived experience, which may not be captured in any published evidence but could steer future research.

The strengths of this review are in its systematic design aiming to incorporate all relevant studies to answer the research question and in its engagement with a stakeholder group to steer the review and engage research with practice. The involvement of a sample of stakeholders raises the potential for biases to be introduced by selection of stakeholders with particular views, opinions or experiences. The use of replicable and transparent systematic review methods helps to minimise this risk. The research question was broad. This limited the search strategy in incorporating all possible terms to address the breadth of the research question and this may mean that some evidence was not found. Another limitation is that the secondary outcomes were only included where they were found in papers that also encompassed the primary outcome. This likely means that the associations found are underestimated. Including any paper with the secondary outcomes would have led to an unmanageable number of papers and, given that the focus of the review was health outcomes, it steered the decision on only including secondary outcomes where relevant in understanding any temporal dynamics to the relationship under study. The grouping of child development measures, used for data synthesis, could be seen as both a strength and a weakness: a strength in that it allowed for the classification of a range of child development measures into developmental domains, and a limitation in that it was a subjective process and, as such, is open to critique.

## 5. Conclusions

Positive socio-emotional development at school starting age appears particularly important for subsequent mental health and weight in adolescence. There are gaps in the evidence about what factors affect the relationships between child development and subsequent health, in particular, the effect of socio-economic factors. More collaborative research across health and education is needed to develop and define appropriate measures of child development across key domains of child development, and also on the relationships and mechanisms between domains of development, particularly cognitive, language and communication, and physical development, and later health, in the context of socio-economic inequality. This requires the design of public health research that respects agency but more clearly theorizes children within their social and economic context [84], so as to encapsulate socio-political, cultural and familial environments. Research designed using longitudinal cohorts could be one way forward here and be considered in future work on this topic. This theoretically informed research and knowledge is imperative to inform interventions to address health inequalities in mid-childhood and adolescence.

## Figures and Tables

**Figure 1 ijerph-18-11613-f001:**
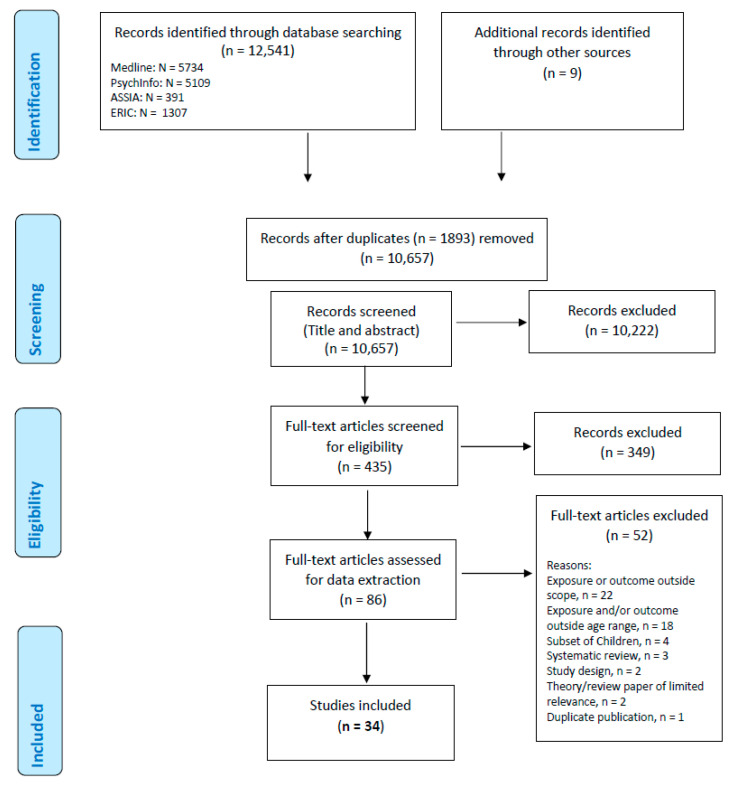
PRISMA flowchart of study selection process.

**Figure 2 ijerph-18-11613-f002:**
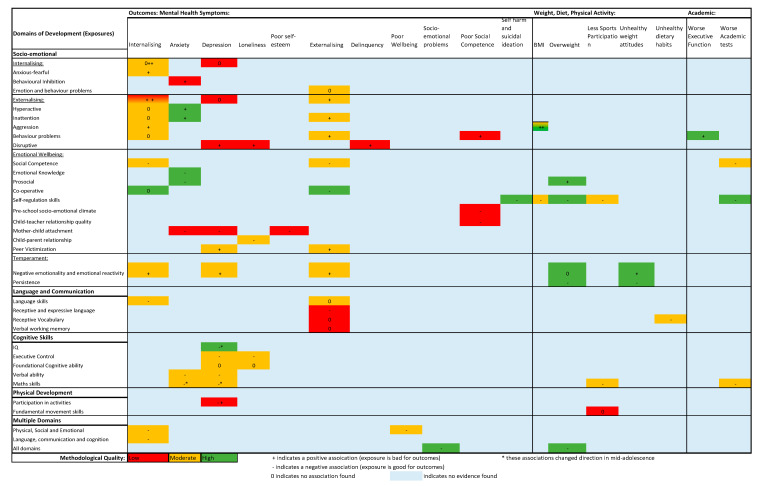
Evidence of associations between domains of “child development” (exposures) and outcomes of mental health symptoms, weight and academic.

**Figure 3 ijerph-18-11613-f003:**
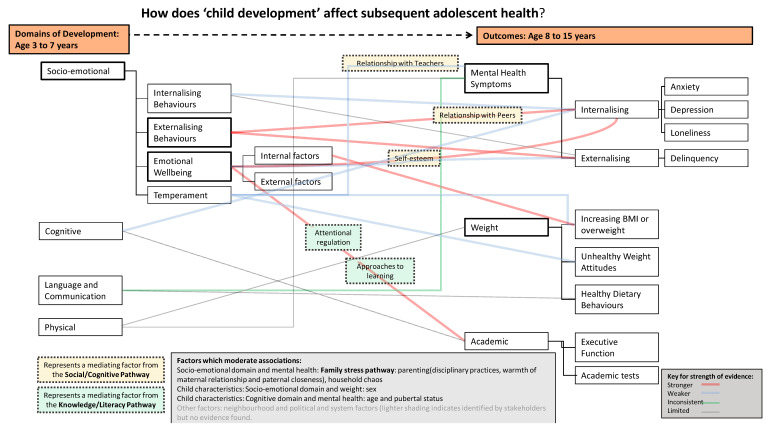
Diagram of the relationship between domains of child development and adolescent outcomes.

**Table 1 ijerph-18-11613-t001:** Summary of eligibility criteria.

	Inclusion	Exclusion
Population and context	Studies must include children, some or all of whom are aged between 3 and 15 years, across socio-economic strata in high-income country settings, defined as OECD membership.	Studies of children from non-OECD countries. Studies which focus solely on a particular subset of children with a particular health or development need.
Exposure	A measure of child development at school starting age (3–7 years), defined as: cognitive or physical or linguistic or socio-emotional development at school starting age, measured by any of the following: School readiness, as measured by scales such as the Bracken Basic Concepts Scale Revised (BBCS-R) [25]. Cognitive development as measured by, for example, non-reading intelligence tests, vocabulary tests, mathematics tests or parent/teacher ratings. Language and literacy (as measured by academic achievement test scores such as pre-reading/reading, vocabulary, oral comprehension, phonological awareness, pre-writing/writing or verbal skills. Emotional well-being and social competence (as measured by behavioural assessments of social interaction, problem behaviours, social skills and competencies, child-parent relationship/child-teacher relationship). Physical development. Studies that explore socio-economic factors which affect associations between child development at school starting age and these outcomes. Studies that explore mechanisms or pathways between child development at school starting age and these outcomes.	Studies reporting neither data nor mechanism between exposure and outcome will be excluded.
**Outcome**	Primary Outcome(s) The review will incorporate evidence health and wellbeing outcomes, reported between the ages of 8 and 15 years, specifically: Weight (BMI). Mental Health (as measured by standard questionnaires or clinically). Socio-emotional behaviour. Proxy measures such as dietary habits and behaviour and measures of wellbeing will be included. Secondary Outcome(s) Performance at the end of primary school (age 10–11), measured by standardized tests. Proxy measures such as executive function.	Studies reporting neither data nor mechanism between exposure and outcome will be excluded.
**Study design and sources**	Observational studies (ecological, case-control, cohort (prospective and retrospective)) RCTs, Quasi-experimental, Review level studies including theory papers.	Cross-sectional studies, conference abstracts, books, dissertations, or opinion pieces.

## Data Availability

Lists of included and excluded articles are available in the published article and the associated Appendix A.

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
