# Peer review of "Relationships between Child Development at School Entry and Adolescent Health—A Participatory Systematic Review"

_ijerph, 2021, doi:10.3390/ijerph182111613_

Round 1

Reviewer 1 Report

Dear authors,

I believe that the manuscript can be improved with the following recommendations:

  1. TITLE OF THE STUDY should be short, simple, easy to understand and between 15 and 20 words in length.
  2. The ABSTRACT is somewhat excessive; it should be within 300 words and under the following subheadings. So that the article can be easily followed by the readers.
  3. On the other hand, it is missing to cite what are the applications of this study: where this study can be useful, give name of area, disciplines, etc.
  4. I miss highlighting the novelty/originality of this study: what is new in this study that can benefit readers and how it is advancing existing knowledge or creating new knowledge on this topic.
  5. INTRODUCTION is correct. It provides the background of the study in simple words and the authors discuss the research problem in very clear words.
  6. METHODOLOGY: For systematic literature review, the questions are clearly formulated, method of collection is correct, as well as the nature of the review.
  7. RESULTS: data are presented in a clear or organised way, supported by appropriate figures and tables.
  8. DISCUSSION: It is recommended that the authors add a discussion section in which they express more clearly the combination of their findings in relation to those previously identified in the literature review, and place them within the context of the theoretical framework underpinning the study. On the other hand, it is appreciated that the limitations of the study are reported.
  9. CONCLUSIONS: It is necessary to report on future lines of research taking into account any limitations of the study mentioned previously.

Author Response

Dear Reviewer, 

Thank you for your considered review. Please find our responses to your points here:

1. TITLE OF THE STUDY should be short, simple, easy to understand and between 15 and 20 words in length.

Response to point 1: Thank you. We have amended the title to: ‘Relationships between child development at school entry and adolescent health – a participatory systematic review’

2. The ABSTRACT is somewhat excessive; it should be within 300 words and under the following subheadings. So that the article can be easily followed by the readers.

Response to point 2: The abstract has been revised and is now within the word count. It is written as per the headings but the headings are not stated (as per the journal submission guidelines).

3. On the other hand, it is missing to cite what are the applications of this study: where this study can be useful, give name of area, disciplines, etc.

Response to point 3: The concluding part of the abstract has been revised to incorporate this point about where the study can be useful. We have added  “More collaborative research across health and education is needed on other domains of development and on the mechanisms that link development and later health, and how any relationship is modified by socio-economic context.” – This is in line with the main conclusions section of the paper.

4. I miss highlighting the novelty/originality of this study: what is new in this study that can benefit readers and how it is advancing existing knowledge or creating new knowledge on this topic.

Response to point 4: It is difficult to cover this within the abstract given the word count.  However, it is covered in the introduction and discussion sections. In the introduction, lines 128-134, in particular we state “…as far as we are aware no review has analysed the evidence on relationships between different dimensions of child development and adolescent health outcomes or assessed the factors which may shape the relationships.” We also highlight in the discussion that this study creates new knowledge by providing “a classification of 4 domains of child development (which) adds to the literature, providing a framework for other researchers to use and to critique.” 

5. INTRODUCTION is correct. It provides the background of the study in simple words and the authors discuss the research problem in very clear words.

6. METHODOLOGY: For systematic literature review, the questions are clearly formulated, method of collection is correct, as well as the nature of the review.

7. RESULTS: data are presented in a clear or organised way, supported by appropriate figures and tables.

Response to points 5, 6 and 7: Thank you

8. DISCUSSION: It is recommended that the authors add a discussion section in which they express more clearly the combination of their findings in relation to those previously identified in the literature review, and place them within the context of the theoretical framework underpinning the study. On the other hand, it is appreciated that the limitations of the study are reported.

Response to point 8: We have added an additional paragraph to the discussion to address these points – see paragraph 2 of the discussion, line 802-812.

9. CONCLUSIONS: It is necessary to report on future lines of research taking into account any limitations of the study mentioned previously.

Response to point 9: We have amended the conclusion to make clearer the future lines of research. See lines 969 to 982.

Reviewer 2 Report

Although the relationship between child development and adolescent health is a long-term concern, it is not easy to clarify the complicated relationship between them. This study uses a literature review method to try to unravel this veil. This work has made an obvious academic contribution. But after all, this is an academic paper. Based on academic requirements, I put forward the following review comments.

  1. In “Abstract”, its content is too long. Abstract is based on the principle of simplification, unnecessary numbers can be deleted, leaving the most important part. This way readers can clearly understand the author's purpose and contribution in this article.
  1. In line 390-392, the authors omitted Table 3 and Table 4.
  1. In line 617, the authors omitted punctuation.
  1. In line 775, the description of the picture should be placed below the picture. Please correct it.
  1. Figure 2 is obviously the most important conclusion of this manuscript. However, from the content of the article, it can be found that several relationships are "limited". For these uncertain relationships, I suggest that the authors strengthen their discussion. Authors can even point out future research suggestions in the "Conclusions" section. Because for a review article, in addition to a comprehensive summary, it is important to find gaps in future research. Therefore, it is recommended that the authors can make appropriate supplements.

Author Response

Dear Reviewer, 

Thank you for your considered review. Please find our responses to your points here:

  1. In “Abstract”, its content is too long. Abstract is based on the principle of simplification, unnecessary numbers can be deleted, leaving the most important part. This way readers can clearly understand the author's purpose and contribution in this article.

Response to point 1: The abstract has been revised and is now within the word count.

  1. In line 390-392, the authors omitted Table 3 and Table 4.

Response to point 2: Typo corrected in the heading for table 4. Table 4 is attached as a png file.

  1. In line 617, the authors omitted punctuation.

Response to point 3: Corrected.

  1. In line 775, the description of the picture should be placed below the picture. Please correct it.

Response to point 4: Corrected.

  1. Figure 2 is obviously the most important conclusion of this manuscript. However, from the content of the article, it can be found that several relationships are "limited". For these uncertain relationships, I suggest that the authors strengthen their discussion. Authors can even point out future research suggestions in the "Conclusions" section. Because for a review article, in addition to a comprehensive summary, it is important to find gaps in future research. Therefore, it is recommended that the authors can make appropriate supplements.

Response to point 5: We have amended the conclusion to state the particular gaps in evidence which require further research and the type of research required. Lines 968-982.

Reviewer 3 Report

The reviewed article is perfectly prepared in every analyzed dimension. The very interesting methodology of conducting literary studies deserves special praise, it can certainly serve as a model in conducting analyzes of a similar type.

I congratulate the authors of the results of the work, reading this work is a great pleasure and intellectual experience.

Author Response

Thank you.

Reviewer 4 Report

Dear authors,

this systematic review is very interesting. 

I have nothing to comment on. 
I think the manuscript is well written, accurate, clear and complete in summarizing the relationship between child development and subsequent adolescent health and identifying factors that influence the relationship.

I would only change the keywords (i.e., child development; adolescent health), replacing them with other words not in the title. This might make it easier to find your manuscript through search engines.

Author Response

Thank you. The key words have been amended.